# Targeted Transposition of Minicircle DNA Using Single-Chain Antibody Conjugated Cyclodextrin-Modified Poly (Propylene Imine) Nanocarriers

**DOI:** 10.3390/cancers14081925

**Published:** 2022-04-11

**Authors:** Willi Jugel, Stefanie Tietze, Jennifer Daeg, Dietmar Appelhans, Felix Broghammer, Achim Aigner, Michael Karimov, Gabriele Schackert, Achim Temme

**Affiliations:** 1Department of Neurosurgery, Section Experimental Neurosurgery and Tumor Immunology, University Hospital Carl Gustav Carus, TU Dresden, Fetscherstraße 74, 01307 Dresden, Germany; willi.jugel@uniklinikum-dresden.de (W.J.); stefanie.tietze@uniklinikum-dresden.de (S.T.); felix.broghammer@uniklinikum-dresden.de (F.B.); gabriele.schackert@uniklinikum-dresden.de (G.S.); 2Leibniz Institute of Polymer Research Dresden e.V., Mailbox 120411, 01069 Dresden, Germany; jenniferdaeg@web.de (J.D.); applhans@ipfdd.de (D.A.); 3Rudolf-Boehm-Institute for Pharmacology and Toxicology, Clinical Pharmacology, Faculty of Medicine, University of Leipzig, 04107 Leipzig, Germany; achim.aigner@medizin.uni-leipzig.de (A.A.); michael.karimov@medizin.uni-leipzig.de (M.K.); 4German Cancer Consortium (DKTK), Partner Site Dresden, 01307 Dresden, Germany; 5German Cancer Research Center (DKFZ), 69120 Heidelberg, Germany; 6National Center for Tumor Diseases (NCT), 01307 Dresden, Germany

**Keywords:** poly(propylene imine), β-cyclodextrin, DNA delivery, p53

## Abstract

**Simple Summary:**

Gene therapy for the treatment of malignancies is an emerging and promising area of particular interest. Therefore, it is imperative to develop effective delivery systems for the specific transfer of nucleic acids into tumors. In comparison to viral vectors, non-viral delivery systems tend to be simple to manufacture, show lower immunogenicity, and are associated with fewer regulatory issues when translated into clinical settings. The aim of our study was to develop single-chain antibody conjugated cyclodextrin-modified poly(propylene imine) (PPI) nanocarriers, comprising β-cyclodextrin-modified PPI, mono-biotinylated, maltose-modified PPI, neutravidin and mono-biotinylated prostate stem cell antigen (PSCA)-specific single-chain antibodies for the targeted transposition of minicircle DNA into PSCA-positive tumor cells. Remarkably, we achieved long-term expression of a therapeutic p53 gene in PSCA-positive tumor cells by combining our tumor-specific hybrid polyplexes with the Sleeping Beauty transposon system in minicircle format.

**Abstract:**

Among non-viral vectors, cationic polymers, such as poly(propylene imine) (PPI), play a prominent role in nucleic acid delivery. However, limitations of polycationic polymer-based DNA delivery systems are (i) insufficient target specificity, (ii) unsatisfactory transgene expression, and (iii) undesired transfer of therapeutic DNA into non-target cells. We developed single-chain antibody fragment (scFv)-directed hybrid polyplexes for targeted gene therapy of prostate stem cell antigen (PSCA)-positive tumors. Besides mono-biotinylated PSCA-specific single-chain antibodies (scFv(AM1-P-BAP)) conjugated to neutravidin, the hybrid polyplexes comprise β-cyclodextrin-modified PPI as well as biotin/maltose-modified PPI as carriers for minicircle DNAs encoding for Sleeping Beauty transposase and a transposon encoding the gene of interest. The PSCA-specific hybrid polyplexes efficiently delivered a GFP gene in PSCA-positive tumor cells, whereas control hybrid polyplexes showed low gene transfer efficiency. In an experimental gene therapy approach, targeted transposition of a codon-optimized p53 into p53-deficient HCT116^p53−/−/PSCA^ cells demonstrated decreased clonogenic survival when compared to mock controls. Noteworthily, p53 transposition in PTEN-deficient H4^PSCA^ glioma cells caused nearly complete loss of clonogenic survival. These results demonstrate the feasibility of combining tumor-targeting hybrid polyplexes and Sleeping Beauty gene transposition, which, due to the modular design, can be extended to other target genes and tumor entities.

## 1. Introduction

Gene therapy for the treatment of malignancies remains an area of particular interest. Previous attempts to deliver gene-coding DNA using replication-deficient viral vectors showed only limited success in preclinical and clinical studies, and have also raised serious safety concerns [1,2]. In comparison, non-viral delivery systems tend to be simple to manufacture, show lower immunogenicity, and are associated with fewer regulatory issues when translated into clinical settings. However, the broad application of therapeutic DNA using non-viral delivery systems is hampered by non-specific toxic side-effects, poor pharmacokinetics, low DNA transfection efficiency, and in most cases only transient gene expression (for review, see [3]).

Among non-viral gene transfer systems, poly(propylene imine) (PPI) and its surface-modified derivatives have emerged as promising DNA carriers. The high rate of positively charged amino groups on the surface of the PPI dendrimers enables electrostatic interaction with the negatively charged DNA [4] and results in the formation of compact nano-sized particles, designated “dendriplexes” [5]. Tuning PPI by surface modifications with for instance poly(ethylene glycol) (PEG) reduces cytotoxicity and inhibits intermolecular aggregation. In addition, surface modifications provide a hydrophilic shell that avoids interaction with the reticuloendothelial system and prolongs circulation time in the bloodstream [6]. Recently, we showed that surface charge shielding by maltose modification of peripheral amino groups greatly improves the biocompatibility of PPI glycodendrimers (mal19-PPI) while simultaneously reducing transfection efficiency. However, upon bioconjugation of mono-biotinylated single-chain antibodies (scFv-P-BAP) for targeting, PPI glycodendrimers, referred to as “polyplexes”, became competent for delivering siRNA to target cells in vitro and in vivo expressing the cognate antigen/receptor [7,8].

We sought to exploit this polyplex system for targeted delivery of gene-encoding therapeutic DNA. As a target on cancer cells, we chose the prostate stem cell antigen (PSCA), a glycophosphatidylinositol (GPI)-anchored tumor-associated antigen which is found in a variety of urogenital-related cancers [9,10,11,12,13], and various other malignancies such as pancreatic adenocarcinoma [14] and glioblastoma [15]. Yet, preliminary studies using targeted polyplexes containing mal19-PPI revealed selective but only moderate transfection efficiency of DNA in target cells, which is likely related to the attenuated endosomal escape of the DNA payload. To improve DNA transfection efficiency of polyplexes, we developed a modified polyplex system, designated hybrid polyplex, combining transfection-incompetent antibody-conjugated mal19-PPIs with β-cyclodextrin-modified PPIs (CD-PPI). β-cyclodextrins are natural cyclic oligosaccharides with seven glucose units in their structure linked by α-(1,4) glucoside bonds [16]. The donut-shaped β-cyclodextrins are characterized by a hydrophilic outer surface coated with hydroxyl groups and a hydrophobic inner cavity coated with ether groups of anomeric oxygen atoms [17]. Noteworthily, β-cyclodextrins are approved by the FDA as excipients in pharmaceutical products [18] that, in addition to their role as solubilizers, stabilizers of colloids, or modifiers for controlled release, can enhance the permeation of drugs through biological membranes (for review, see [19]).

To achieve sustained transgene expression of the transgene, we flanked its expression cassette with Sleeping Beauty (SB) inverted terminal repeats (ITRs); these SB transposons are mobile genetic elements that efficiently integrate DNA into the genome. To enable transposition, we employed simultaneous delivery of DNA encoding a hyperactive form of SB transposase (SB100X). In targeted cells, the transposase recognizes the ITRs of the transposon and enables genomic integration via a cut-and-paste mechanism [20,21].

A major concern when considering the clinical application of gene-coding DNA delivery is the non-specific cytotoxic effects on cells and enhanced degradation of introduced foreign DNA caused by unmethylated CG dinucleotides located in the bacterial backbone of plasmid DNA [22]. To address this safety concern and to optimize packaging of DNA encoding genes into polyplexes, we genetically engineered the content and size of plasmids. More specifically, the transposon as well as the expression cassettes for SB100X were flanked by attB and attP recognition sites for bacterial PhiC31 integrase to produce minicircle (MC) DNA devoid of bacterial backbones [22].

To this end, PSCA-specific hybrid polyplexes efficiently delivered MC encoding a GFP transposon in PSCA-positive 293T cells, whereas control hybrid polyplexes conjugated to non-specific control scFv-P-BAPs showed low gene transfer efficiency. In an experimental gene therapy approach, delivery of MC encoding a TP53 transposon by PSCA-specific hybrid polyplexes into p53-deficient HCT116^p53−/−/PSCA^ colon cancer cells and PTEN-deficient H4^PSCA^ glioma cells led to decreased clonogenic survival. Remarkably, surviving glioma clones failed to proliferate further. Interestingly, a notable number of colon cancer clones that escaped transgenic p53 showed loss of transgene expression. Noteworthily, in such colon cancer cells, transgenic p53 expression could be induced by treatment with the DNA-damaging antibiotic zeocin. In summary, our results demonstrate the feasibility of combining tumor-targeting hybrid polyplexes and Sleeping Beauty gene transposition for gene therapy, which due to the modular design can be extended to other target genes and tumor entities.

## 2. Materials and Methods

### 2.1. Synthesis of PPIs

Synthesis of 4th generation maltose-modified PPI dendrimers (mal19-PPI (G4) 14,900 g/mol) and biotinylated mal19-PPIs (mal19-PPI-biotin (G4), 15,490 g/mol) were described previously [7,8]. For synthesis of β-cyclodextrin-modified PPIs, the complete reactions were carried out under an argon protection atmosphere. In a first reaction flask (heated, degassed, and filled with argon), 2nd and 4th generation PPI dendrimers were added and degassed for around 1 h under high vacuum, followed by the addition of 2 mL of degassed anhydrous DMSO and triethylamine (Et3N). In a second reaction flask (heated, degassed, and filled with argon) βCD-PEG-CO_2_H and BOP were dissolved in 2 mL of degassed anhydrous DMSO. Subsequently, the resulting reaction mixture was stirred for 2 h. PPI dendrimer solution was slowly added to the activated βCD-PEG-CO_2_H solution, stirred for 2–2.5 days at room temperature, and dialyzed for 2 days in 5 L water (membrane tube with MWCO 2000 g/mol) with exchanging aqueous solution. After the freeze-drying, a viscous liquid was obtained. Used quantities for CD-PPI (G2) and CD-PPI (G4) synthesis are presented in Appendix B (Table A1) and results of ^1^H NMR characterization are presented in Appendix B as well (Figure A1, Figure A2, Figure A3, Figure A4 and Figure A5), including the synthesis and NMR characterization of βCD-PEG-CO_2_H.

### 2.2. Cell Lines

The hypodiploid colorectal carcinoma-derived HCT116 cells with homozygous knockout of p53, designated HCT116^p53−/−^ (kindly provided by B. Vogelstein, Johns Hopkins University, Baltimore, U.S.), the near triploid (+/−3n) glioma cell line H4, and the human embryonic kidney cell lines 293T, 293T^PSCA^, and 293T^huBirA^ have been described previously [7,23,24,25]. HCT116^p53−/−/PSCA^ cells with ectopic expression of PSCA were generated by lentiviral transduction of HCT116^p53−/−^ cells using the lentiviral p6NST53-PSCA vector, described previously [26], followed by geneticin (Invitrogen, Waltham, MA, USA) selection. Packaging of viral particles and transduction were performed using a three vector system described previously [27]. HCT116^p53−/−^ and HCT116^p53−/−/PSCA^ cells were maintained in RPMI-1640 completed with 10% *v/v* heat-inactivated FCS, 2 mM L-glutamine, 100 µg/mL streptomycin, 100 U/mL penicillin, and 10 mM HEPES (all from Life Technologies, Carlsbad, CA, USA). H4, H4^PSCA^, 293T, and 293T^PSCA^ cell lines were cultured in DMEM completed with 4.5 g/L glucose, 10% *v*/*v* heat-inactivated FCS, 100 U/mL penicillin, 100 μg/mL streptomycin, and 10 mM HEPES (all from Life Technologies). 293T^huBirA^ cells were maintained in DMEM complete supplemented with 50 μM Biotin-C6 (Sigma-Aldrich, St. Louis, MO, USA) for scFv production. Cells were cultured at 37 °C with 5% CO_2_ in a humidified incubator. All cell lines were authenticated (Multiplexion, Heidelberg, Germany).

### 2.3. Electrophoretic Mobility Gel Shift Assay

MC-DNA (1 µg) was incubated for 30 min at room temperature with increasing amounts of PPI or CD-PPI corresponding to mass ratios of 1:5 to 1:0.2. The dendriplexes were then separated by agarose gel electrophoresis [1% (*w*/*v*)] and analyzed under UV light (G:Box Chemi XX9).

### 2.4. Measurement of Cell Viability

2 × 10^4^ 293T^PSCA^ cells were plated in 96-well plates and grown in 200 μL DMEM medium until 70% confluency, before adding different concentrations of PPI(G2), PPI(G4), mal19-PPI, CD-PPI (G2), or CD-PPI (G4). After 24 h, all wells of the assay were incubated with 20 μL AlamarBlue solution (Thermo Fisher Scientific, Waltham, MA, USA) for an additional 4 h. For normalization, untreated cells were included as the negative control. Cells lysed with 5% Triton X-100 (Sigma-Aldrich) served as the positive control. Finally, the fluorescence intensities of the reduced AlamarBlue in the wells were detected by a fluorescence imaging system (Synergy 2, BioTek, Winooski, VT, USA) and 560EX nm/590EM nm filter settings. The cytotoxicity of PPI glycodendrimers on cells was calculated in relation to untreated controls, which was set to 100% viability.

### 2.5. Plasmids for Minicircle Production and Sleeping Beauty Transposition

To generate a minicircle transposon encoding GFP, the synthetic SB transposase restriction sites IR/DR(L) and IR/DR(R) were ligated to the corresponding restriction sites SmaI—ClaI and StuI—EcoRV of the parental minicircle vector pMC.CMV-GFP (System Biosciences, Palo Alto, CA, USA).

A synthetic codon-optimized cDNA encoding the full 393 amino acids of p53 (Eurofins MWG Biotech, Ebersberg, Germany) fused to a T2A endoproteolytic cleavage site and a puromycin resistance gene was ligated to the corresponding restriction sites ClaI and HindIII of the parental minicircle vector pMC.CMV (System Biosciences), resulting in pMC-p53-puroR. In pMC-p53-puroR, the p53/puroR transgene was flanked by transposase restriction sites IR/DR(L) and IR/DR(R). pMC-puroR lacking the p53 coding sequence was used as mock control. The pCMV(CAT)T7-SB100 (Addgene, Watertown, MA, USA, plasmid # 34879) encoding hyperactive SB100X Sleeping Beauty transposase has been described previously [28]. The SB100X gene sequence was amplified by PCR adding XbaI restriction sites and ligated to the XbaI restriction sites in the MCS of the parental minicircle vector pMC.CMV-MCS (System Biosciences). All vector inserts were confirmed by sequencing (Microsynth Seqlab, Göttingen, Germany).

### 2.6. Production of scFv-P-BAP

The DNA sequence and features of the single-chain antibody-derivative scFv(AM1)-P-BAP have been described previously [8,23]. The construct includes an N terminal Igκ leader sequence, a biotin acceptor peptide (P-BAP), and a C-terminal c-myc epitope and a 6x histidine (His)-tag. The biotinylated scFvs were expressed in transiently transfected 293T^huBirA^ producer cells and purified from the harvested cell culture supernatant by Ni2+-NTA affinity chromatography and Avidin-biotin affinity chromatography as described previously.

Briefly, the harvested medium was spun down and clarified supernatant was passed through a Ni2+-NTA spin column (Qiagen, Hilden, Germany) and subsequently washed with PBS containing 150 mM NaCl and 10 mM/20 mM imidazole. To elute bound scFv-P-BAPs the column was loaded with 500 μL 1x PBS containing 150 mM NaCl and 350 mM imidazole. Eluted scFv-P-BAPs were dialyzed in PBS twice for 2 h and for 12 h at 4 °C. Biotinylated scFv-P-BAPs were further purified using monomeric avidin affinity chromatography (Thermo Fisher Scientific, Waltham, MA, USA) according to the protocol of the manufacturer. Eluted scFv-P-BAPs were subsequently dialyzed again, as described previously. When needed, the recombinant scFv-P-BAPs were concentrated by employing Ultra-15 Amicon tubes (Merck Millipore, Burlington, VT, USA) and stored in aliquots at −20 °C.

### 2.7. Production of Minicircles

Minicircles were produced using the MC-Easy™ Minicircle DNA Production Kit (System Biosciences) according to the manufacturer’s protocol. Briefly, pMC-GFP, pMC-puroR, pMC-p53-puroR, and pMC-SB100 were grown in E. coli bacterial strain ZYCY10P3S2T harboring an arabinose-inducible system for simultaneous expression of PhiC31 integrase and Sce-I endonuclease. After incubation with induction medium, intramolecular (cis-) recombination generated MC from the parental plasmid mediated by PhiC31 integrase. The remaining parental plasmid-DNA backbone was degraded by Sce-I endonuclease. MC-GFP (3.7 kb), MC-puroR (2.6 kb), MC-p53-puroR (3.7 kb), and MC-SB100 (4.5 kb) were purified from the medium using Plasmid Plus Maxi Kit (Qiagen) according to the manufacturer’s protocol.

### 2.8. Assembly of Tumor-Specific Hybrid Polyplexes and Targeted Transfection of Cells

For the assembly of tumor-specific hybrid polyplexes, a two-step conjugation protocol was used. First, CD-PPIs were mixed with MC at a mass ratio of 5:1, resulting in bioconjugation adduct 1. In parallel, neutravidin (Thermo Fisher Scientific), scFv-P-BAP, and mal19-PPI-biotin were incubated for 30 min at room temperature at a molar ratio of 2:1:1, resulting in bioconjugation adduct 2. After saturation of the remaining free biotin binding sites of neutravidin with 0.3 mM D-biotin (Sigma-Aldrich), the intermediate conjugates were mixed for 30 min at room temperature and subsequently used for further studies. Hybrid polyplexes with 1 µg MC-DNA contained 100 pmol scFv-P-BAP, 50 pmol neutravidin, and 50 pmol mal19-PPI-biotin. Twenty-four hours before transfection, the indicated amounts of target cells were plated in a medium containing heat-inactivated FCS and treated with 6.5–7.5 × 10^3^ hybrid polyplex nanoparticles/cell for 12 h before exchanging with a fresh medium.

### 2.9. Multiparameter Nanoparticle Tracking Analysis

Multiparameter nanoparticle tracking analysis was performed using the ZetaView^®^ PMX120 (Particle Metrix, Inning am Ammersee, Germany) according to the manufacturer’s instructions to determine the particle quantities, zeta potential, and particle size of scFv(AM1)-P-BAP-hybrid polyplexes with MC. Hybrid polyplexes were prepared as described above. Data were analyzed using the manufacturer’s software (ZetaView 8.05.05).

### 2.10. Western Blot Analysis

The produced scFv-P-BAP and scFv-P-BAP:neutravidin conjugates were investigated using immunoblot analysis. Therefore, 1 μg of scFv-P-BAP was separated by SDS PAGE (12% polyacrylamide gel) under reducing conditions. Neutravidin conjugates including 1 μg of scFv-P-BAP were separated under non-reducing conditions. Subsequently, separated proteins were transferred by semi-dry Western Blot to a PVDF membrane (Whatman plc, Maidstone, UK). The PVDF membrane was blocked with 5% non-fat dry milk in Tris-buffered saline (TBS) containing 0.1% Tween 20 (TBS-T) during a 1 h incubation followed by washing with TBS-T and TBS. Immune detection was performed using a primary monoclonal murine anti-c-myc antibody (1:5000, Invitrogen) and a secondary polyclonal rabbit anti-mouse IgG HRP conjugate (1:1000; Dako Agilent, Santa Clara, CA, USA). Biotinylated scFv-P-BAPs were detected by HRP-conjugated anti-biotin antibody (1:3000, Sigma-Aldrich). For immunoblot analysis of the conjugates, scFv(AM1)-P-BAP, and neutravidin were mixed in advanced at various molar ratios from 8:1 to 0.5:1 in 1 × PBS and incubated for 30 min at room temperature.

For analysis of targeted TP53 transposition, HCT116^p53−/−/PSCA^ cell clones were pooled after transfection with scFv(AM1)-P-BAP-guided hybrid polyplexes, containing MC-SB100 and MC-p53-puroR or MC-puroR as a mock control, grown to 80% confluence in DMEM complete containing 2 µg/mL puromycin and incubated with 500 µg/mL zeocin (Invitrogen) for 4 h. Zeocin-treated and-non-treated cells were lysed in lysis buffer (10 mM Tris-HCl; pH 8.0; 140 mM NaCl; 1% Triton X-100). Equal amounts of total protein were subjected to SDS-PAGE under reducing conditions and blotted on a PVDF membrane using semi-dry Western Blotting. After blocking PVDF membrane with 5% BSA in TBS-T, p53, phospho-p53 (Ser 15) and p21^waf/cip^ were detected with a polyclonal rabbit-anti-human p53 antibody (1:500; 7F5; Cell Signaling Technology, Danvers, MA, USA), a polyclonal rabbit-anti-human phospho-p53 antibody (1:500; ab1431; Abcam, Cambridge, UK) and a polyclonal rabbit-anti-human p21 antibody (1:500; 12D1; Cell Signaling Technology), followed by an HRP-conjugated anti-rabbit IgG secondary antibody (1:1000; Dako Agilent). To detect equal loading, PVDF membranes were stripped and subsequently stained with an anti-α tubulin antibody (1:5000; Sigma-Aldrich), followed by a secondary polyclonal rabbit anti-mouse IgG HRP conjugate (1:1000; Dako Agilent). Visual capturing of proteins was performed by a chemiluminescent method using Luminata Forte Western HRP substrate (Merck Millipore) and the G:Box Chemi XX9 (VWR, Radnor, PA, USA) gel doc documentation system. Analysis of immunoblots was performed using Fiji software (ImageJ 1.51 k, National Institute of Health, Bethesda, MD, USA).

### 2.11. Polymerase Chain Reaction

Analysis of transposon integration was performed by polymerase chain reaction (PCR) using PhusionTM High-Fidelity DNA Polymerase (Thermo Fisher Scientific). 1 × 10^5^ 293T cells polyethylenimine (PEI)-transfected with a total amount of 2 µg DNA containing MC-transposon-GFP alone or MC-transposon-GFP together with MC-SB at a molar ratio of 3:1, were harvested 28 days after transfection. Genomic DNA was prepared using the QIAamp DNA Mini Kit (Qiagen). Primers for the transposon (T) were MC-GFP-inside-For 5′-ccaacaagatgaagagcacc-3′ and MC-GFP-inside-Rev 5′-aagggacgtagcagaaggac-3′, for the minicircle backbone (MC) MC-GFP-outside-For 5′-gacggcgacaagcaaacatg-3′ and MC-GFP-outside-Rev 5′-tcgccttctatcgccttcttg-3′ and for the transposase SB100X (SB) MC-SB100X-For 5′-gtctggttcatccttgggag-3′ and MC-SB100X-Rev 5′-gggtcattgtcgtgttggaag-3′. The amplification protocol was: 98 °C denaturation 30 s, followed by 35 cycles at 98 °C denaturation 10 s, 59 °C annealing 30 s and 72 °C extension 90 s 5 µL PCR-product were separated by agarose gel electrophoresis [1% (*w*/*v*)] and analyzed under UV light (G:Box Chemi XX9, Syngene, Cambridge, UK).

### 2.12. Clonogenic Survival Assay

For analysis of direct effects after targeted p53 transposition, 2 × 10^4^ HCT116^p53−/−PSCA^ or H4^PSCA^ cells plated in 6-well plates in DMEM complete were incubated with scFv(AM1)-P-BAP MC-SB100/MC-p53-puroR hybrid polyplexes. Cells treated with scFv(AM1)-P-BAP MC-SB100/MC-puroR hybrid polyplexes were included as mock control. Treated cells were continuously grown in a medium containing 2 µg/mL puromycin starting 24 h after transfection. Clonogenic survival was analyzed after 14 days. In order to investigate the long-term replicative potential of the treated cells, parallel experiments were performed and surviving HCT116^p53−/−PSCA^ and H4^PSCA^ cell clones were pooled. 1 × 10^3^ cells thereof were plated on 10 cm dishes and analyzed after 14 days or grown until 80% confluency for Western blot analysis of transgenic p53 (see above). In all clonogenic survival experiments, cell clones were stained with crystal violet staining solution (Merck Millipore), photographed and counted using ImageJ software (National Institute of Health, USA).

### 2.13. Flow Cytometry

Binding of biotinylated scFv(AM1)-P-BAP to PSCA-positive target cells and isogenic parental cells was analyzed by flow cytometry. 2 × 10^5^ cells were stained with 5 μg scFv-P-BAP for 1 h at 4 °C, followed by secondary anti-biotin-VioBlue antibody (Miltenyi Biotec, Cologne, Germany). Cells stained with the secondary antibody alone served as the control. To evaluate the targeted delivery of DNA, 2 × 10^5^ 293T^PSCA^ cells were incubated with scFv-P-BAP hybrid polyplexes loaded with Cy3-labelled plasmid DNA (Mirus Bio, Madison, WI, USA, plasmid # MIRUMIR7904, 2.7 kb) for 4 h at 37 °C. Cy3-plasmid-loaded hybrid polyplexes conjugated with control scFv(MR1.1)-P-BAP were included as a negative control. Subsequently, the cells were washed with 0.1% Heparin/PBS (Sigma-Aldrich). To evaluate the transfection efficiency, 1 × 10^5^ 293T^PSCA^ were incubated with PPI:MC-GFP dendriplexes containing 1 µg MC-DNA and PPI at different mass ratios for 48 h. At least 2 × 10^4^ cells were measured by MACSQuant Analyser 10 flow cytometer (Miltenyi Biotec) and analyzed by FlowJo software version 10.1 (TreeStar, Ashland, OR, USA).

### 2.14. Confocal Laser Scanning Microscopy

For visualization of cellular DNA uptake, 6 × 10^5^ 293T^PSCA^ cells grown on a coverslip were incubated with scFv(AM1)-P-BAP hybrid polyplexes loaded with Cy3-labelled plasmid DNA at 37 °C. After 24 h, cells were fixated with 4% paraformaldehyde in PBS (*w*/*v*). Subsequently, cell membranes and nuclei were stained with Alexa Fluor 647-conjugated wheat germ agglutinin (WGA, Life Technologies) and Hoechst 33,342 (Invitrogen). The early endosomes were stained with EEA1 (E-8) Alexa Fluor 647 (Santa Cruz Biotechnology, Dallas, TX, USA) according to the manufacturer’s protocols. The coverslips were placed upside down in a mounting medium (Vector Laboratories, Burlingame, CA, USA) on a microscope slide. Cells treated with Cy3-labelled hybrid polyplexes conjugated with non-binding control scFv(MR1.1)-P-BAP were included as negative control. Fluorescence microscopy images were captured by a confocal laser scanning microscope (Leica SP5, Leica Microsystems, Wetzlar, Germany) and analyzed by Fiji software (ImageJ 1.51k, National Institute of Health).

### 2.15. Statistical Analysis

All experiments were performed at least two times in at least triplicates. Differences between groups were examined for statistical significance using Student’s *t*-test. Values of *p* < 0.05 were considered statistically significant: * *p* < 0.05, ** *p* < 0.01, *** *p* < 0.001.

## 3. Results

### 3.1. Characterization of β-cyclodextrin-Modified PPIs

The toxicity of cationic PPI dendrimers is a major concern, especially when applying them as DNA carriers for cancer therapy. Therefore, we assessed the cell viabilities of 293T cells incubated with increasing concentrations of 2nd generation PPI (G2), 4th generation PPI (G4), and with the corresponding β-cyclodextrin-modified PPIs (PPI (G2) modified with 25% β-cyclodextrin and PPI (G4) modified with 6% β-cyclodextrin (CD-PPI (G2), CD-PPI (G4)) (Figure 1A). We also included PPI (G4) modified with 19% maltose (mal19-PPI (G4)) in the experiments. As shown in Figure 1B, the cytotoxicity of PPI dendrimers increased with the PPI generation. PPI (G2) was essentially non-toxic even at a concentration of 10 µM, whereas the LD50 value calculated for PPI (G4) was 3.5 µM. After surface modification of PPIs with β-cyclodextrin, PPI (G2) remained non-toxic; remarkably, the LD50 value of PPI (G4) increased up to 4.75 µM, suggesting that shielding of the peripheral primary amino surface groups by β-cyclodextrin decreased cytotoxicity. This is consistent with the results observed with mal19-PPI (G4), where, as described previously, grafting maltose units onto the surface groups significantly reduced cytotoxicity. To confirm dendriplex formation with minicircle DNA (MC), unmodified and β-cyclodextrin modified PPIs were incubated at MC/PPI mass ratios ranging from 1:5 to 1:0.2. As analyzed by gel electrophoresis, MC mobility was completely inhibited regardless of PPI surface modification at MC/PPI mass ratios 1:5, 1:3 and 1:1, indicating that the positive net charge of the PPIs was sufficient for MC complexation (Figure 1C). Treatment of 293T cells with PPI dendriplexes containing GFP-encoding MC at mass ratios ranging from 1:1 to 10:1 revealed that β-cyclodextrin modification strongly correlated with increased transfection efficiency. Surface modification with β-cyclodextrin resulted in the mean in a 35-fold or 3.5-fold increase in transfection efficiency for CD-PPI (G2) or (G4), respectively, using mass ratio of 5:1, compared with the corresponding unmodified PPIs. Best transfection efficiencies were obtained with CD-PPI (G2) at a PPI/MC mass ratio of 5:1. As described previously, unmodified PPI dendrimers and mal19-PPI were nearly transfection incompetent [7]. As positive control, the cells were transfected with increasing PEI/MC mass ratios (Figure 1D,E).

### 3.2. Characterization of scFv-P-BAP

All scFv-P-BAP constructs contained an N-terminal Igκ chain leader sequence for extracellular secretion as well as a C-terminal c-myc-epitope and a 6x histidine (His)-tag for detection and purification, respectively (Figure 2A). In addition to the detection of the c myc-epitope in Western Blot analysis, the successful enzymatic biotinylation of scFv(AM1)-P-BAP and of the control antibody scFv(MR1.1)-P-BAP was confirmed by a biotin-specific antibody (Figure 2B). The observed bands show a molecular mass of approximately 55 kDa, which roughly fits the calculated mass. The Coomassie Brilliant Blue stained polyacrylamide gel confirmed purity and successful expression of the full-length proteins (Figure 2C). In subsequent experiments, the binding properties of scFv(AM1)-P-BAP were assessed. As shown in Figure 2D, the results demonstrated highly efficient binding to the PSCA-positive target cells 293T^PSCA^, HCT116^p53−/−PSCA^ and H4^PSCA^. Detection of scFv(AM1)-P-BAP via a biotin-specific secondary antibody confirmed the accessibility of the biotin residue under native conditions. In comparison, scFv(AM1)-P-BAP did not bind to PSCA-negative isogenic control cells 293T, HCT116^p53−/−^ and H4. As expected, scFv(MR1.1)-P-BAP, which binds to the neo-epitope of the mutated epidermal growth factor variant III (EGFRvIII) did not bind to PSCA-positive target cells and therefore represented a valid negative control scFv for further experiments.

### 3.3. Characterization of scFv-Guided Hybrid Polyplexes

The stable conjugation of scFv(AM1)-P-BAP to neutravidin was validated by Western Blot analysis shown in Figure 3A. Therefore, the two components were incubated at molar ratios of scFv-P-BAP to neutravidin of 8:1, 4:1, 3:1, 2:1, 1:1, and 0.5:1. The conjugates remained stable during SDS-PAGE, which is due to the high affinity of neutravidin to biotin. The scFv(AM1)-P-BAP/neutravidin conjugates were detected by their C-terminal c-myc-epitope of the scFv-P-BAP. Distinct scFv(AM1)-P-BAP/neutravidin conjugate bands were detected at approximately 100 and 160 kDa for molar scFv-P-BAP/neutravidin ratios 2:1, 1:1, and 0.5:1. Non-conjugated scFv(AM1)-P-BAP proteins containing an accessible biotin residue were detected by an anti-biotin antibody at approximately 55 kDa at molar scFv/neutravidin ratios 8:1, 4:1, and 3:1. In contrast, non-conjugated biotinylated scFv(AM1)-P-BAP proteins were absent when using scFv-P-BAP/neutravidin molar ratios of 2:1, 1:1, and 0.5:1 (Figure 3A).

For generation of tumor-specific hybrid polyplexes loaded with gene-coding DNA, neutravidin was conjugated with mono-biotinylated scFv-P-BAP and mono-biotinylated mal19-PPI-biotin in a 1:2:1 molar ratio. To avoid potential biotin-driven agglutination due to undesired cross-linking of the components, the remaining biotin-binding pockets of neutravidin were saturated by the addition of D-biotin after conjugation of scFv-P-BAPs and mal19-PPI-biotin. Complexation between CD-PPI (G2) or (G4) and MC was achieved at a mass ratio of 5:1. The electrostatic interactions between the intermediate conjugates mal19-PPI-biotin/neutravidin/scFv-P-BAP (bioconjugation adduct (1) and CD-PPI/MC (bioconjugation adduct (2) resulted in scFv(AM1)-P-BAP-guided hybrid polyplexes (Figure 3B). Assessment of the physiochemical properties revealed a negative net charge for the intermediate conjugate mal19-PPI-biotin/neutravidin/scFv-P-BAP (−30.8 ± 0.5 mV) and a positive net charge for CD-PPI (G2) or (G4) dendriplexes (5.3 ± 0.2 mV, 3.7 ± 0 mV). Complexation between both intermediate conjugates resulted in a mean negative net charge of −16.8 ± 0.2 mV and −11.7 ± 0.3 mV and a mean particle size of 146.5 ± 72.9 nm and 136.9 ± 55.1 nm for scFv(AM1)-guided hybrid polyplexes containing CD-PPI (G2) or (G4), respectively (Figure 3C).

To analyze targeted DNA delivery to tumor cells, 293T^PSCA^ were treated with MC-GFP-loaded scFv(AM1)-P-BAP-guided hybrid polyplexes. As negative control, non-specific hybrid polyplexes containing the EGFRvIII-specific scFv(MR1.1)-P-BAP were included. Up to 87.2 ± 2.8% and 64.5 ± 4.7% 293T^PSCA^ cells appeared GFP-positive after treatment with scFv(AM1)-guided hybrid polyplexes containing CD-PPI (G2) and (G4), respectively. Remarkably, transfection efficiencies decreased to 38.6 ± 4.9% and 10% ± 0.9 for hybrid polyplexes containing CD-PPI (G2) or CD-PPI (G4) after conjugation of the control antibody scFv(MR1.1)-P-BAP (Figure 3D,E). These results clearly indicate that hybrid polyplexes, when coupled with scFv-BAPs, can successfully deliver gene-coding DNA in target cells expressing the cognate surface receptor. In contrast, transfection efficiencies of MC-GFP-transposon loaded scFv(AM1)-P-BAP- and non-specific scFv(MR1)-P-BAP polyplexes containing mal19-PPI (G4) were 38.5 ± 4.5% and 14.0 ± 1.1%, respectively, indicating selective but only moderate transfection efficiency of DNA in target cells (Appendix A).

### 3.4. Targeted Delivery of DNA in PSCA-Positive Target Cells Employing scFv(AM1)-P-BAP Hybrid Polyplexes

Further, we focused on the route of internalization of scFv-guided hybrid polyplexes, particularly scFv(AM1)-guided hybrid polyplexes containing CD-PPI (G4), which confers most specific transfection efficiencies in 293T^PSCA^ cells. To this end, 293T^PSCA^ were treated with scFv(AM1)-P-BAP-guided hybrid polyplexes loaded with Cy3-labelled plasmid DNA. As negative control, non-specific hybrid polyplexes containing the EGFRvIII-specific scFv(MR1.1)-P-BAP were included. Flow cytometry analysis demonstrated scFv(AM1)-P-BAP-mediated internalization of the hybrid polyplexes by 293T^PSCA^ cells, whereas significantly less Cy3 signals were detectable in cells after treatment with non-specific hybrid polyplexes conjugated with scFv(MR1.1)-P-BAP (Figure 4A,C). Confocal laser scanning microscopy studies, shown in Figure 4B,D, supported the data obtained by flow cytometry analysis. The tumor-specific scFv(AM1)-P-BAP hybrid polyplexes were found to be internalized by 293T^PSCA^ cells. Staining with a wheat germ agglutinin conjugate revealed that the Cy3-labelled plasmid DNA was located intracellularly in the cytoplasm. Additional staining with the early endosome associated protein EEA1 (early endosome antigen (1) demonstrated partial location in early endosomes. In contrast, the non-specific scFv(MR1.1)-P-BAP-guided hybrid polyplexes showed less nanoparticle uptake. Quantitative analysis of Cy3 signals revealed significantly more signals overall and per cell after treatment with scFv(AM1)-P-BAP hybrid polyplexes compared to control hybrid polyplexes (Figure 4E,F).

To investigate if a receptor-mediated endocytosis contributes to uptake of scFv-P-BAP hybrid polyplexes, we blocked clathrin-dependent and clathrin-independent endocytosis using chlorpromazine [29] and filipin III [30]. Both filipin III and chlorpromazine treatment of cells lead to significant decrease in the fraction of GFP-positive 293T^PSCA^ cells when transfected with scFv(AM1)-P-BAP hybrid polyplexes loaded with MC encoding GFP. None of the used endocytosis inhibitors completely blocked transfection of GFP, suggesting that besides clathrin-dependent and -independent endocytic pathways a direct uptake mechanisms of scFv-P-BAP-guided hybrid polyplexes might have supported the delivery of transposon MCs encoding GFP (Appendix A).

### 3.5. Targeted Transposition of Minicircle DNA Using scFv-P-BAP-Guided Hybrid Polyplexes

To achieve stable gene transfer, we sought to combine targeted DNA-minicircle delivery using scFv-P-BAP-guided hybrid polyplexes with the Sleeping Beauty transposon system, which enables transposition-based integration of DNA-sequences into chromosomes. So far, efficiency and functionality of the SB transposon system was confirmed in 293T cells transfected with MC encoding GFP transposon with or without co-delivery of minicircle encoding SB transposase SB100X (MC-SB). As expected, control cells transfected with only MC encoding transposon-GFP lost GFP expression over time whereas cells transfected with transposon and SB100X showed enduring gene expression in approximately 50% of cells. (Appendix A). In line with this, PCR analysis of cell lysates revealed stable integration of the transposon (T) into the genome, but not of minicircle sequences outside the transposon sequence and of the transposase SB100X, respectively (Appendix A).

Next, we extended our studies employing targeted transposition of TP53 in p53-deficient HCT116^p53−/−/PSCA^ and H4^PSCA^ tumor cells. To this end, cells were treated with MC encoding p53-puroR transposon and MC encoding SB. Cells transfected with scFv(AM1)-P-BAP-guided hybrid polyplexes loaded with MC encoding transposon-puroR and MC-SB served as negative control. Whereas the hypodiploid colorectal carcinoma cell line HCT116^p53−/−^ is derived from the parental chromosomally stable cell line HCT116, the near triploid glioma cell line H4 display an increased chromosomal instability [31]. Furthermore, H4 cells have lost one copy of the TP53 gene and are further characterized by loss of PTEN, which eventually results in enhanced MDM2-mediated ubiquitinylation and degradation of p53 protein [25,32]. Theoretically, H4 cells are more prone to p53 gene therapy than HCT116^p53−/−^ cells since continuous mitotic defects and DNA-damage should result in posttranslational modifications (i.e., phosphorylation, acetylation) and stabilization of transgenic p53. Analysis of direct effects 10 days after targeted transposition of TP53 revealed decreased clonogenic survival of HCT116^p53−/−/PSCA^ transfected with p53-transposon when compared to mock controls. Yet, the effect of targeted p53 transposition was even more increased in transfected H4^PSCA^ glioma cells when compared to the control (Figure 5A). Surviving H4 glioma cell clones had in general fewer cell numbers (data not shown) and pooled clones could not be further propagated. In contrast, pooled HCT116^p53−/−/PSCA^ cell clones that survived scFv(AM1)-P-BAP-guided transposition of p53 were readily propagated onto new culture dishes (Figure 5B). This intriguing result prompted us to investigate p53 levels in surviving p53 transposon-treated HCT116^p53−/−/PSCA^ cells. We anticipate that less DNA damage occurs during mitosis of chromosomally stable HCT116^p53−/−/PSCA^ cells limiting the effects of p53. As depicted in Figure 5, transgenic p53 protein was not detected in cell lysates of cells treated with p53-transposon, which might be related to its continuous degradation. Strikingly, by treatment with DNA-damaging reagent zeocin, we were able to stabilize transgenic p53 protein expression. As depicted in Figure 5C, treatment of surviving HCT116^p53−/−/PSCA^ cells carrying the p53 transposon with zeocin readily induced a DNA-damage response indicated by upregulation of p53 protein and its phosphorylated form phospho-p53 (Ser15). Furthermore, expression of p53’s downstream transcriptional target p21^waf/cip^ was noted (Figure 5C). In controls cells, either treated with PBS and/or carrying a mock-transposon, p53 expression was undetectable and no induction of p21^waf/cip^ was observed, which indicates continuous degradation of transgenic p53 in HCT116^p53−/−/PSCA^ cells carrying p53 transposon. Altogether, our results demonstrate that scFv-P-BAP-directed hybrid polyplexes in combination with the SB transposon system can efficiently deliver transposons to target cells displaying a cognate surface antigen.

## 4. Discussion

Successful therapeutic gene delivery depends on the vector used for overcoming the main obstacles to DNA transfer: low uptake by the plasma membrane and insufficient release of DNA molecules into the cytoplasm. The vectors currently used can be roughly divided into viral vectors and non-viral vectors. Non-viral techniques for gene delivery are direct physical methods such as microinjection and particle bombardment, i.e., gene gun, electroporation, sonoporation, laser beam, and magnetofection, and the chemical-based approaches such as non-viral carrier systems (liposomes, lipoplexes, polymers, peptides, nanoparticles; for a review, see [33]). Despite the great advantages of non-viral vectors—they are neither immunogenic nor carcinogenic and can provide large quantities of therapeutic DNA efficiently and cost-effectively—none of these vectors has yet proven more efficient than viral vectors for the transfer of therapeutic genetic material.

Recently, we established a novel tumor-specific siRNA delivery system, consisting of maltose-modified PPI/PPI-biotin, neutravidin, and a biotinylated scFv for targeted siRNA delivery to tumor cells [7]. Yet, subsequent studies using polyplexes containing mal19-PPI revealed selective but only modest transfection efficiency of gene-coding DNA into corresponding target cells. In the present study, we therefore further modified this modular platform technology by combining transfection-incompetent antibody-conjugated mal19-PPIs with transfection-competent β-cyclodextrin-modified PPIs (CD-PPI) to reduce cytotoxicity of PPI while increasing transfection efficiency. It is well known that dendritic poly(amido amine) dendrimers with unshielded cationic surface groups exhibit generation-dependent high cytotoxicity in vitro. Moreover, the number of free amine groups is linearly related to their cytotoxicity [34,35]. Recently, we showed that partial surface modifications by direct maltose coupling to PPI-dendrimers are an effective way to optimize cytotoxic profiles [7]. We obtained comparable results using partial surface modifications with β-cyclodextrin. Remarkably, in contrast to maltose-modified PPIs, β-cyclodextrin-modified PPIs showed high non-specific transfection efficiency of gene-coding DNA. This is consistent with previous reports describing the incorporation of β-cyclodextrin into polycationic dendrimer vectors for enhanced DNA transfer [36]. Yet, the assembly of hybrid polyplexes consisting of β-cyclodextrin-modified PPI, maltose-modified PPI-biotin, neutravidin, and a mono-biotinylated scFv significantly reduced the non-specific transfection efficiency. Since a positive surface charge has been described as a prerequisite for efficient in vitro transfection [37,38], we hypothesize that the reduced zeta potential of the PSCA-specific hybrid polyplexes compared to dendriplexes containing only β-cyclodextrin-modified PPIs prevents non-specific in vitro transfection. Consequently, targeted gene delivery to PSCA-positive target cells with PSCA-specific hybrid polyplexes was mediated by the biotinylated scFv-P-BAP via receptor-mediated uptake. We chose PSCA as a target antigen on tumor cells, since it is found on a broad range of tumor entities [13,14,39,40]. Recent studies from our group have established PSCA as a targetable antigen, which after receptor-crosslinking by anti-PSCA immunoconjugates induces endocytosis [23]. In this study, we identify a mixed clathrin- and caveolae-mediated uptake as the mechanism of PSCA-specific hybrid polyplex internalization. These results are similar to our recent report, which revealed a mixed clathrin- and caveolae-mediated uptake of nanoparticle-like immunoconjugates comprising mono-biotinylated anti-PSCA-scFv conjugated via neutravidin to mono-biotinylated Toll-like receptor 3 (TLR3) agonist [23]. However, when using hybrid poylplexes, it cannot be completely ruled out that their β-cyclodextrin moieties, when brought in close proximity to the cell membrane, augment direct uptake of hybrid polyplexes.

In our concept of hybrid polyplexes for targeted gene therapy, we furthermore sought to avoid delivery of bacterial non-methylated CpG-motifs and to simultaneously implement a transposable element for stable expression of the delivered transgene. This combination should prevent unwanted activation of Toll-like receptor 9 or other pattern recognition receptors in treated cells, eventually leading to inflammation and activation of defense mechanisms, which ultimately lead to degradation of foreign DNA [41]. In this regard, it has been shown that electroporation of eukaryotic cells with SB transposase and transposon in minicircle format leads to 20-fold higher DNA transfer efficiency compared to conventional plasmids, and to profoundly reduced cellular toxicity in human cells by up to 50% [42]. Of note, our concept using minicircle Sleeping Beauty transposons goes beyond a recent concept using biodegradable poly(β-amino ester)-based nanoparticles electrostatically complexed with polyglutamatic acid-modified anti murine CD3ε-(Fab)2 for targeted delivery of full length plasmids encoding PiggyBac transposase and chimeric antigen receptors (CARs) transposon to murine CD3ε -positive cells. In that respect, disappointing ex vivo transfection efficiencies in murine T cells might be related to the above mentioned defense mechanisms [43]. Yet, this study and our own results clearly indicate the feasibility of transposon systems to achieve long-lasting gene expression. In particular, clonogenic survival assays of chromosomally stable HCT116^p53−/−/PSCA^ colorectal cancer cells and chromosomally instable H4^PSCA^ glioma cells demonstrated significant decreased cell survival in response to targeted transposition of TP53 in vitro using our PSCA-specific hybrid polyplexes. While the small amount of surviving glioma cells clones could not be propagated further in cell culture, a notable fraction of colon cancer clones that escaped transgenic p53 showed loss of p53 protein expression and was further passaged in cell culture. Remarkably, in these cells, profound transgenic p53 expression was restored by treatment with the DNA-damaging bleomycin family antibiotic zeocin, which demonstrates efficiency of the transposon system and furthermore suggests that cancer cells with stable karyotype are also a legitimate target for p53 gene therapy.

## 5. Conclusions

In summary, we successfully developed a modular platform technology for the targeted delivery of therapeutic gene-coding DNA to tumor cells. By combining our tumor-specific hybrid polyplexes with the Sleeping Beauty transposon system in minicircle format, we achieved long-term transgenic p53 gene replacement in PSCA-positive tumor cells. In further experiments, the reproducibility for other gene therapy targets needs and feasibility in pre-clinical in vivo models must be investigated to pave the way for potential clinical applications.

## 6. Patents

W.J., S.T., D.A. and A.T. hold IP rights for the hybrid-polyplex system for targeted delivery of DNA to eukaryotic cells.

## Figures and Tables

**Figure 1 cancers-14-01925-f001:**
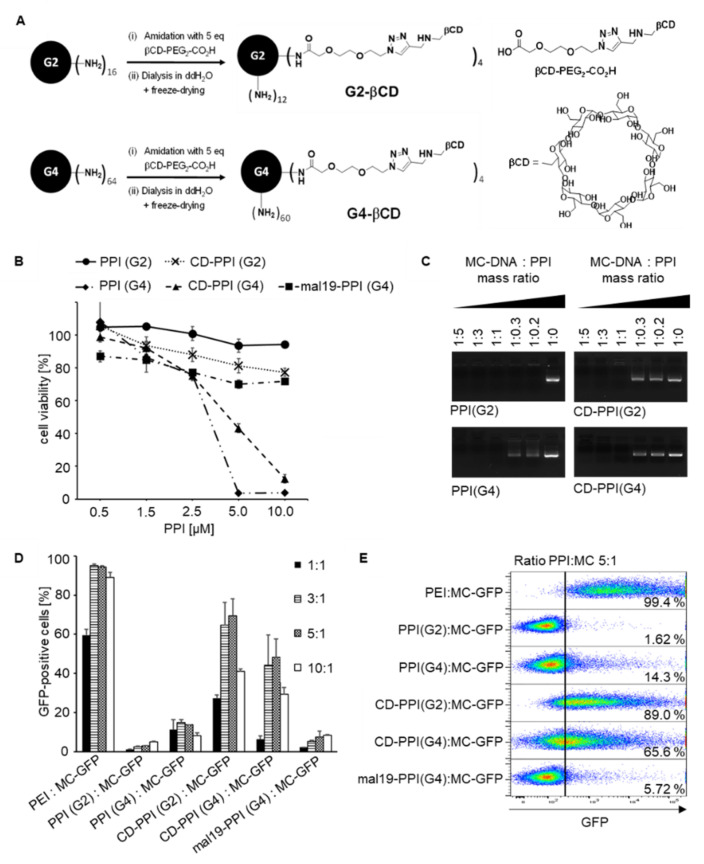
Properties of surface-modified β-cyclodextrin PPIs. (**A**): Chemical structure of β-cyclodextrin modified PPIs. (**B**): Cytotoxicity profile of PPIs with various grades of surface modification with β-cyclodextrin or maltose (2nd generation PPI (PPI (G2)), 4th generation PPI (PPI(G4)), PPI (G2) modified with 25% β-cyclodextrin (CD-PPI (G2)), PPI (G4) modified with 6% β-cyclodextrin (CD-PPI (G4) and PPI (G4) modified with 19% maltose (mal19-PPI (G4)). 293T cell were incubated with increasing concentrations of PPIs and measured in an AlamarBlue assay (*n* = 3, mean ± SD). (**C**): Electrophoretic mobility gel shift assay of minicircle (MC) DNA binding to PPI (G2), PPI (G4), CD-PPI (G2), and CD-PPI (G4). (**D**): Transfection efficiency of unmodified and corresponding β-cyclodextrin-modified PPIs. 293T cells were incubated with increasing ratios of PPI:MC-GFP and analyzed by flow cytometry. As controls, cells were transfected with PEI:MC-GFP and mal19-PPI (G4):MC-GFP (*n* = 3, mean ± SD). (**E**): Representative Dot Plot analysis on the transfection efficiency of unmodified and corresponding β-cyclodextrin-modified PPIs. 293T cells were transfected with PPI:MC-GFP at a mass ratio of 5:1. As controls, cells were transfected with PEI:MC-GFP and mal19-PPI:MC-GFP in the same ratio.

**Figure 2 cancers-14-01925-f002:**
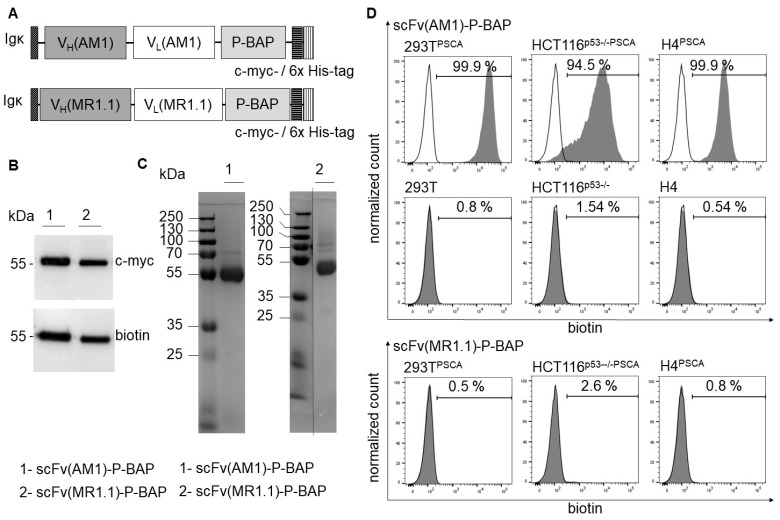
Production and characterization of recombinant biotinylated scFv(AM1)-P-BAP and scFv(MR1.1)-P-BAP. (**A**): Schematic presentation of the scFv(AM1)-P-BAP and control antibody scFv(MR1.1)-P-BAP protein domains. (**B**): Western Blot analysis of biotinylated scFv(AM1)-P-BAP and scFv(MR1.1)-P-BAP using anti-c-myc and anti-biotin antibodies. Uncropped Western Blots can be found at Appendix A. (**C**): Coomassie Brilliant Blue-stained polyacrylamide gel of purified scFv(AM1)-P-BAP and scFv(MR1.1)-P-BAP recombinant antibody derivatives. (**D**): Flow cytometry analysis of 293T, HCT116^p53−/−^, H4, 293T^PSCA^, HCT116^p53−/−PSCA^ and H4^PSCA^ cells stained with scFv(AM1)-P-BAP or scFv(MR1.1)-P-BAP. Binding of the scFvs was detected by secondary anti-biotin-VioBlue (grey histograms). Open histograms represent control staining using only a secondary antibody.

**Figure 3 cancers-14-01925-f003:**
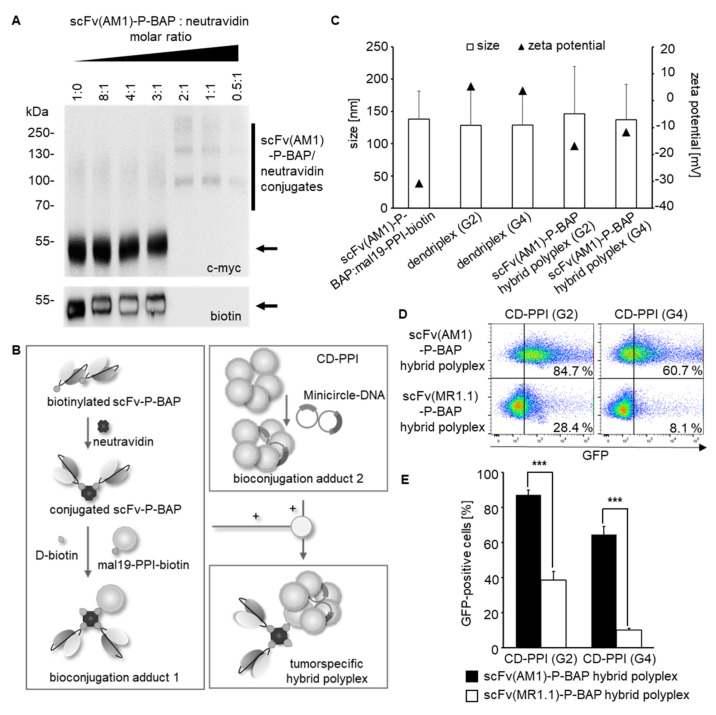
Assembly of scFv(AM1)-P-BAP-guided hybrid polyplexes. (**A**): Titration for the binding of scFv(AM1)-P-BAP to neutravidin with increasing molar ratios. Western Blot analysis showing scFv(AM1)-P-BAP/neutravidin complex formation or unbound scFv(AM1)-P-BAP using anti-c-myc and anti-biotin antibodies. Uncropped Western Blots can be found at Appendix A. (**B**): Schematic representation of the successive conjugation of scFv(AM1)-P-BAP-guided hybrid polyplexes. (**C**): Assessment of particle sizes and zeta potential of scFv(AM1)-P-BAP-guided hybrid polyplexes. (**D**): Representative dot plot analysis of 293T^PSCA^ cells, transfected with MC-GFP using scFv(AM1)-P-BAP-guided hybrid polyplexes in comparison to PSCA-unspecific scFv(MR1.1)-P-BAP-hybrid polyplexes. (**E**): Transfection efficiency of scFv(AM1)-P-BAP-guided hybrid polyplexes in comparison to PSCA-unspecific scFv(MR1.1)-P-BAP-hybrid polyplexes. (*n* = 3, mean ± SD). *** *p* < 0.001.

**Figure 4 cancers-14-01925-f004:**
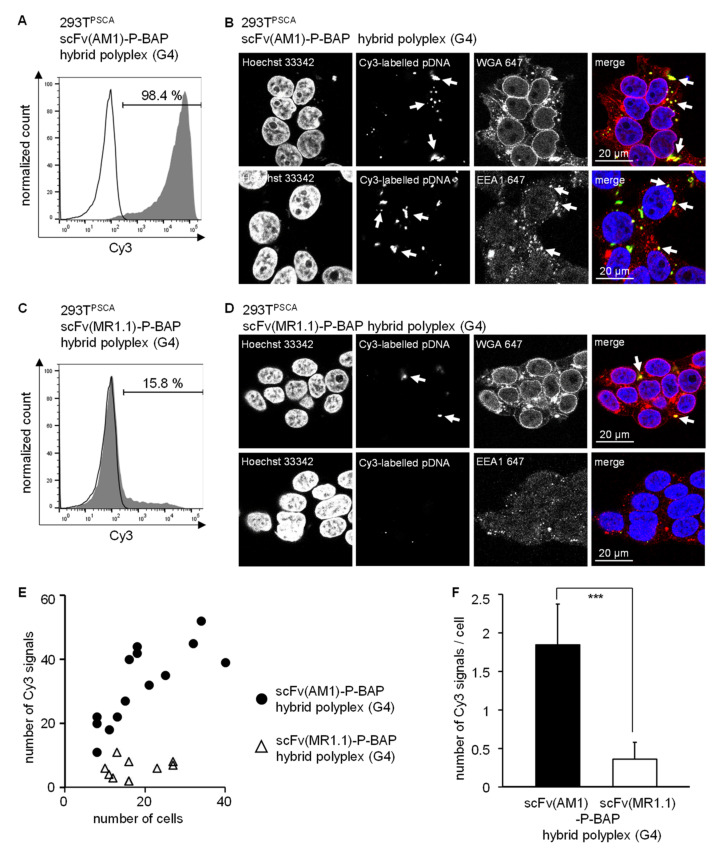
Targeted delivery of scFv(AM1)-P-BAP-guided hybrid polyplexes (G4) in PSCA-positive cells. (**A**,**C**): 293T^PSCA^ cells were treated with scFv(AM1)-P-BAP or scFv(MR1.1)-P-BAP hybrid polyplexes containing Cy3-labelled plasmid DNA for 4 h (grey histograms). As control, untreated 293T^PSCA^ cells were utilized (open histograms). After Heparin-washing of surface-bound antibodies, the internalized Cy3-labelled plasmid DNA was measured by flow cytometry. (**B**,**D**): Confocal laser scanning microscopy analysis of 293T^PSCA^ cells treated with scFv(AM1)-P-BAP or scFv(MR1.1)-P-BAP hybrid polyplexes containing Cy3-labelled plasmid DNA. To visualize the route of internalisation, cells were additionally stained with wheat germ agglutinin or early endosomal marker EEA1. Arrows depict Cy3-labelled pDNA or early endosomes. (**E**,**F**): Quantification of Cy3-labelled pDNA dots per image section or per cell. At least 14 image sections were analyzed (mean ± SD). *** *p* < 0.001.

**Figure 5 cancers-14-01925-f005:**
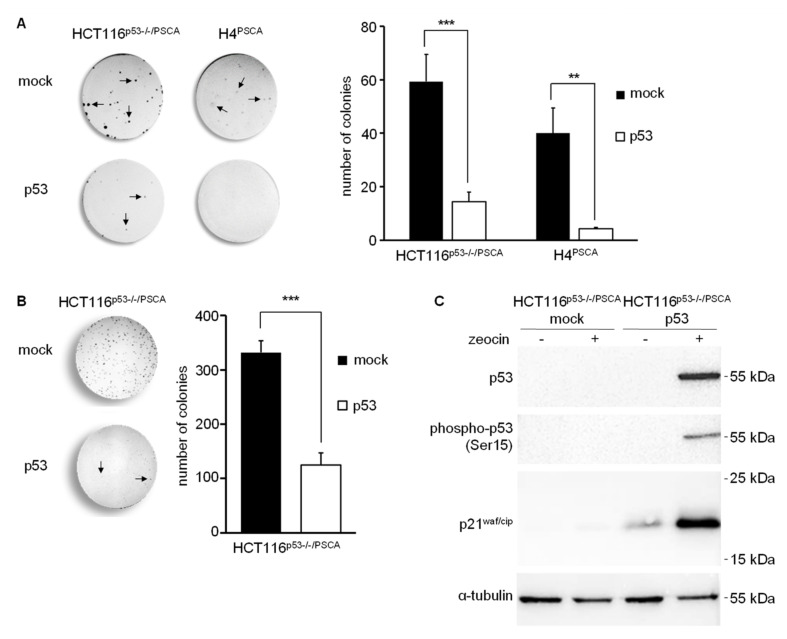
scFv(AM1)-P-BAP-guided transposition of TP53 in p53-deficient, PSCA-positive tumor cells. (**A**): Clonogenic assay of A: HCT116^p53−/−PSCA^ colorectal carcinoma cells or H4^PSCA^ glioma cells treated with scFv(AM1)-P-BAP MC-p53-puroR hybrid polyplexes and cultured in DMEM complete supplemented with 2 µg/mL puromycin or (**B**): Surviving HCT116^p53−/−PSCA^ colorectal carcinoma cell clones that escaped transgenic p53. Cells were cultured for 10 days before staining with crystal violet. The relative number of colonies with more than 15 cells were counted (*n* = 3, mean ± SD). Cells transfected with scFv(AM1)-P-BAP MC-puroR hybrid polyplexes served as negative control (mock). (**C**): Western blot analysis of HCT116^p53−/−PSCA^ colorectal carcinoma cells surviving the targeted transfection with p53 transposon demonstrates an increase in steady state protein levels of p53, phospho-p53 (Ser15) and p21^waf/cip^ in response to DNA damage by the bleomycin family antibiotic zeocin when compared to untreated cells. Cells with targeted transfection of MC-puroR and therefore devoid of p53 transgene served as negative control. Uncropped Western Blots can be found in Appendix A. ** *p* < 0.01, *** *p* < 0.001.

## Data Availability

All data are available upon request to the corresponding author.

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
