# Peer review of "Targeted Transposition of Minicircle DNA Using Single-Chain Antibody Conjugated Cyclodextrin-Modified Poly (Propylene Imine) Nanocarriers"

_cancers, 2022, doi:10.3390/cancers14081925_

Round 1

Reviewer 1 Report

The authors have extensively presented their research work on the antibody-conjugated nano carriers  for gene therapy. All methods and results align moderately well and have been discussed in an organized manner. The manuscript is scientifically sound, however, a thorough grammar check would be required for some minor spelling and grammatical errors.

Author Response

We are thankful for the advices from reviewer 1. We have carried out a grammar check.

Reviewer 2 Report

Very interesting work that has a very well-established background on all presented levels, ie nanoparticle technology, active tumor targeting, the SB system, etc. A few minor points:

-Why 5h incubation with AlamarBlue? Protocol is 1-4h.
-What do the authors suggest of the unknown 1H NMR signals?
-Why are G2 modified with 31% and G4 with 7% CD?
-Why is the mal19‐PPI:MC system referred to as a control for endocytosis?
-Why was the mass ratio used to express the mc/ppi analogies? Molar or even charge ratio would be more informative.
-A mass ratio of 5:1 seems too high in mc to achieve complexation. Is this attributed to the low zeta potential of cd-ppis?
-What is the zeta potential of cd-ppi/mc, since this is conjugated with the adduct 1?

-Line 23 PSCA should be explained.
-Line 79 polyplex.
-PEI is not explained anywhere.
-Line 643 remove "to".

Author Response

Why 5h incubation with AlamrBlue?

It was a mistake on our part in the method section. Thank you for your deep proofreading. We corrected it in the manuscript.

What do the authors suggest of the unknown 1H NMR signals?

This 1H NMR signal is also provided when the precursor βCD-PEG-CO2H is synthesized and is remained in the final product of CD-PPI (G2) and CD-PPI (G4) (not indicated, but with enlargement of signals one can find it as well). We assume that this signal belongs to the cyclodextrin ring, but cannot really prove it due to non-existing correlation to other 1H NMR signals. Therefore, we are not able to suggest something to the unknown 1H NMR signal.

Why are G2 modified with 31% and G4 with 7% CD?

Due to the same substitution degree of 4 for CD and different number of amino groups in both generations, thus both generations outline different percent in CD modification. The percentage of G2 and G4 is corrected in the text with 25 and 6 %, respectively, due only 4 CD units in CD-PPI (G2) with 16 amino groups and CD-PPI (G4) with 64 amino groups. This point is also corrected in Figure 1. The right degree of substitution was given in the SI, but was not corrected in the main text by the submission process and we regret this point.

Why is mal19-PPI:MC system referred to as a control for endocytosis?

Recently, we established a novel tumor-specific siRNA delivery system, consisting of maltose-modified PPI/PPI-biotin, neutravidin and a biotinylated scFv for targeted siRNA delivery to tumor cells. Subsequent studies using polyplexes containing mal19-PPI revealed selective but only modest transfection efficiency of gene-coding DNA into corresponding target cells. So we used the mal19-PPI:MC system for comparison. Our control for endocytosis was the unspecific MR1.1 hybrid polyplex.

Why was the mass ratio used to express the MC:PPI analogies?

Normally, molar ratio would be better. But we used MC with different sizes. Thus, there are large deviations between the charges of the different MC with the same amount of substance. The majority of the net charge DNA is derived from the negatively charged phosphate groups in the backbone, which are proportional tot he mass.

A mass ratio of 5:1 seems to high in mc to achieve complexation. Is this attributed to the low zeta potential of cd-ppis?

We determinde the complexation and transfection efficiency at different CD-PPI:MC mass ratios. In gel shift assay, we achieved a full complexation at the ratio of 5:1. Additionally we get the best results for transfection with this ratio. We diddn’t investigate the zeta potential of the CD-PPIs, they were to small for the Zetaview technology.

What is the zeta potential of CD-PPI/MC?

We determined the zeta potential of adduct 1(-30.8 mV) and adduct 2 (5.3 mV; 3.7 mV). Adduct 2, called dendriplex, consists of CD-PPI and MC. After conjugation, the whole hybrid polyplex had a net charge of -16.8 mV for G2 and -11.7 mV for G4.

Line 23, 79 and 643 were corrected. PSCA and PEI are explained, now.

Reviewer 3 Report

Manuscript cancers-1650663

In this paper a modular platform technology is developed for the
targeted delivery of therapeutic gene‐coding DNA to tumor cells.

The authors describe in detail the methods and interpret the results comprehensively. Further in-vivo experiments may prove the efficacy of the developed delivery system and its potential for clinical applications.

I recommend acceptance

Author Response

We are thankful for the advices from reviewer 3. We have carried out a grammar check.